# Perspectives on Disability and Non-Communicable Diseases in Low- and Middle-Income Countries, with a Focus on Stroke and Dementia

**DOI:** 10.3390/ijerph16183488

**Published:** 2019-09-19

**Authors:** Josephine E. Prynn, Hannah Kuper

**Affiliations:** 1Faculty of Population Health, University College London, 62 Huntley Street, London WC1E 6DD, UK; josephineprynn@gmail.com; 2International Centre for Evidence in Disability, Clinical Research Department, London School of Hygiene & Tropical Medicine, Keppel Street, London WC1E 7HT, UK

**Keywords:** disability, non-communicable disease, stroke, dementia, rehabilitation, healthcare access

## Abstract

Non-communicable diseases (NCD) and disability are both common, and increasing in magnitude, as a result of population ageing and a shift in disease burden towards chronic conditions. Moreover, disability and NCDs are strongly linked in a two-way association. People living with NCDs may develop impairments, which can cause activity limitations and participation restriction in the absence of supportive personal and environmental factors. In other words, NCDs may lead to disabilities. At the same time, people with disabilities are more vulnerable to NCDs, because of their underlying health condition, and vulnerability to poverty and exclusion from healthcare services. NCD programmes must expand their focus beyond prevention and treatment to incorporate rehabilitation for people living with NCDs, in order to maximize their functioning and well-being. Additionally, access to healthcare needs to be improved for people with disabilities so that they can secure their right to preventive, curative and rehabilitation services. These changes may require new innovations to overcome existing gaps in healthcare capacity, such as an increasing role for mobile technology and task-sharing. This perspective paper discusses these issues, using a particular focus on stroke and dementia in order to clarify these relationships.

## 1. Introduction

Globally, the WHO estimates that there are one billion people with disabilities, and disability may be both a risk factor and consequence of non-communicable diseases (NCDs) [1]. NCDs are diseases which cannot be transmitted between people, and include cardiovascular diseases, cancers, chronic respiratory diseases, multiple sclerosis, and diabetes, among many others. Globally, NCDs are responsible for 73% of deaths [2], and have appropriately gained recognition over the past 20 years as a major global public health concern. The increasing prominence of NCDs as a cause of death globally is partly due to success in reducing communicable disease mortality and maternal and child mortality [2], and therefore in itself is not inherently a problem. What is concerning, however, is that of the 80% of global NCD deaths that occur in LMIC, nearly half are in people less than 70 years old [3]. Moreover, 80% of years lived with disability are now attributable to NCDs [4]. The WHO World NCD Action Plan 2013 set out the target of a 25% relative risk reduction in premature mortality from cardiovascular diseases, cancer, diabetes, and chronic respiratory diseases by 2025, but a focus on functioning among survivors with these conditions is currently lacking [5].

The aim of this perspective paper is to consider disability in relation to NCDs—both in terms of vulnerability of people with disabilities to experiencing NCDs as well as the incorporation of rehabilitation within the NCD response to alleviate disability among those living with NCDs. We will illustrate these discussions with examples from stroke and dementia. Stroke is the second leading cause of death and third leading cause of disability adjusted life years (DALYs) lost worldwide, and is of particular importance in low- and middle-income countries (LMIC) where 70% of strokes and 78% of DALYs lost to stroke occur [6]. Although population-level data on stroke epidemiology are sparse from LMIC, stroke incidence appears to be higher in LMIC than high-income countries (HIC) [6], and age-standardised stroke incidence is increasing in LMIC [6,7]. Dementia is also becoming an increasing concern with an ageing world population: the number of people living with dementia globally more than doubled between 1990 and 2016 [8], and by 2030 it is estimated that 63% of people with dementia will be living in LMIC [9].

## 2. What Is Disability?

It is important to first define what disability is, in order to consider the relationship between disability and NCDs. The WHO’s International Classification of Functioning (ICF) considers disability as a “dynamic interaction between a person’s health condition, environmental factors, and personal factors” [10]. Therefore, a person with a disability is a person with a health condition causing a physical, psychological or cognitive impairment which can then lead to activity limitation and participation restriction, including in work, social, and family life. This process is mediated by personal factors (e.g., age, gender, wealth) and environmental factors (e.g., policy framework, infrastructure accessibility), as shown in Figure 1.

For example, a person who has had a stroke might develop hemiparesis, which is a physical impairment. This impairment could lead to difficulty walking and limit the person’s opportunities to participate in society (e.g., attend church), depending on whether they had access to walking aids, whether their environment was accessible, and whether they were affected by stigma in their community, among multiple other factors, and these may differ by personal factors such as gender and age, shown in Figure 2. Thus, two people with the same physical impairment following stroke may have very different experiences of disability depending on these personal and environmental factors. Equally, a person with a cognitive impairment secondary to dementia may have difficulty interacting with society because of problems understanding or communicating with other people. This social exclusion will be mediated by the support they receive in terms of memory aids and their communication environment, such as whether people they are communicating with use short and easily comprehensible sentences [11].

## 3. What Is the Association between NCDs and Disability?

There is a two-way association between NCDs and disability, as NCDs may lead to disability, and at the same time, people with disabilities may be more vulnerable to NCDs. These different links will be discussed in turn.

### 3.1. NCDs as a Cause of Disability

NCDs can often lead to disabilities. In particular, cancer, cardiovascular disease, chronic respiratory disease and diabetes, the NCDs most targeted by the WHO World NCD Action Plan, contribute substantially to disability worldwide. A systematic review of 105 articles from high income countries showed that the prevalence of difficulties with activities of daily living (i.e., eating, bathing, dressing) was reported to be 10.4–34.5% amongst cancer survivors, 21.1–64.1% in those with cardiovascular disease, 7.4–49.8% in those with chronic respiratory disease and 12.2–54.5% for those with diabetes [12]. This pattern is also observed in LMICs—for instance, a study in India found that among older adults, people with NCDs had higher disability scores, and those with more than one NCD had higher disability scores than those with a single disease [13]. Furthermore, data from the Global Burden of Disease study show that 9 out of the top 10 causes of years lived with disability were NCDs, including low back pain, major depression, neck pain, hearing loss, migraine, diabetes, chronic respiratory disease, anxiety and other musculoskeletal conditions (the only exception was iron-deficiency anaemia) [14].

The mechanisms through which NCDs contribute to disability are highly varied, and include impacts in physical, cognitive and psychological domains. NCDs can lead directly to impairments, which may reduce activities and participation. As an example, diabetes can lead to diabetic eye disease, diabetic neuropathy, and macrovascular complications such as stroke and myocardial infarction [15]. A meta-analysis across 26 studies (only one LMIC included) showed that diabetes is associated with a 50–80% increase in risk of mobility impairment and reductions in activities of daily living [16]. Although data are lacking, it appears that this pattern also exists in LMICs—in rural Malawi, people with diabetes were more likely to report functional difficulties than those without [17]. 

The link between NCDs and disability may also be indirect. Chronic respiratory diseases can lead to functional limitations and dyspnoea, but also to anxiety and depression [12], and cancers can have wide-ranging effects on a patient’s functional status [18]. Aside from the conditions themselves, medication side effects can be drivers of disability, as can stigma related to certain NCDs such as severe mental health disorders [19,20].

Our example conditions of stroke and dementia are also strongly linked to disability. Stroke can cause a wide range of physical impairments such as limb weakness, difficulties with balance, vision, and speech, cognitive impairments [21], and psychological sequelae such as depression [22]. A systematic review found that globally 18–73% of people experienced cognitive impairment after stroke and 15–79% experienced aphasia [12]. These patterns are also noted in LMICs; In Tanzania, over 60% people who had survived stroke were found to have moderate or severe functional difficulties, measured using the Barthel index [23]. These impairments can lead to activity limitations and participation restriction. In urban South Africa, after at least 6 months following a stroke, 43% of people felt that they had achieved no or minimal re-integration with their families, with particular difficulties reported around being unable to take on family responsibilities or return to work [24]. Similar patterns of participation restriction were reported post-stroke in Malawi in 2018 [25]. In Nigeria, a qualitative study examining the experience of life after stroke found that participants reported *“I cannot visit the market as usual; now my outing has been restricted to only once a week”*, *“I am cut-off from social visits”*, and *“the only thing I can do now is to sit and look”* [26]. Similar themes were found in South Africa, Tanzania and Rwanda, with participants reporting social isolation *“I cannot get where others are. I visit nobody… it’s very hard”*, restriction in religious activities *“I go to church less than usual”*, and work *“I am not able to cultivate than I did before”* [27]. Stroke survivors in Nigeria had lower quality of life scores than people without stroke, which particularly reflected difficulties in the social, family, work, and emotional domains [28].

Dementia is also associated with very high levels of disability and is by far the leading NCD cause of disability and care needs in older people worldwide [9]. The Global Burden of Disease report gave dementia a disability weight of 0.67 when calculating associated DALYs, which was higher than almost any other condition in view of the very substantial impact it has on the person living with the condition [9,29]. In addition to memory, dementia affects abilities in language, abstract thought, spatial relations, attention, and judgement, and by definition is severe enough to impact an individual’s function [30]. There is little published on how dementia impacts activities and participation in LMIC, but from HIC, there is ample evidence that dementia can cause difficulty with domestic life activities, interpersonal relationships, and community, social and civic life [31,32].

### 3.2. Vulnerability to NCDs among People Living with Disabilities

Furthermore, disability itself can act as a risk factor for NCDs through several different pathways [33,34]. The underlying health condition leading to disability may have further health impacts. For instance, diabetes can cause visual loss but also raise the risk of cardiovascular disease. People with disabilities may also be vulnerable to health conditions, such as people who are immobile after stroke or dementia and thereby develop pressure sores. Disability is closely linked to poverty [35,36], and poverty is a well-accepted mediator of health, with the most vulnerable in society being at highest risk of health problems [37]. People with disabilities may also have more risky behaviour with respect to NCD risk factors. Evidence from high-income settings suggests that people with disability are more likely to be physically inactive [38,39,40], which is one of the most important risk factors for cardiovascular disease, diabetes, and dementia [41,42,43,44]. In a population-based study in Malawi, people with disabilities were more likely to be obese than people without disabilities [17], and in some populations, it has also been observed that people with disabilities are more likely to smoke [39,40].

People with disabilities are also more likely to have unmet health needs in a wide variety of contexts [45,46,47], which is likely to increase the risk of NCD incidence, progression and mortality. Evidence from high-income settings suggests that people with disabilities are less likely to access cancer-screening, even when free of charge, and the difference remains after adjustment for multiple confounders such as age, socio-economic status, and education level [48]. For instance, in Japan, people with schizophrenia were less likely to attend cancer screening, and this association became stronger with increasing functional difficulties [49]. In the USA, adults with disabilities were less likely to receive preventative healthcare including cancer screening [50]. Barriers to accessing screening services impact on people’s abilities to perceive a health need and seek care, so that health problems may be diagnosed late and have a worse outcome. People with disabilities are also on average less likely to receive treatment when they have an NCD, including anti-hypertensive drugs [51]. While limited, evidence also suggests that when people with disabilities do seek healthcare, they receive poorer quality services, contributing in turn to worse health [1].

### 3.3. Difficulties in Access to Healthcare for People with Disabilities

Levesque conceptualises access to healthcare into the framework shown in Figure 3, whereby difficulties in access to healthcare arise from the side of the healthcare system and the patient and occur along the healthcare seeking journey [52]. This framework highlights that people with disabilities face challenges additional to those faced by people without disabilities at each stage, mostly because of structural deficiencies in how healthcare services are provided.

Transportation to health facilities is a frequently cited barrier to accessing healthcare services. In Chile, people with disabilities were three times more likely to report difficulty getting to health services than people without disabilities [53]. In rural South Africa, people with disabilities reported difficulty using their wheelchairs on roads to the healthcare centre, due to mud, gravel, or uneven surfaces, and difficulty paying for and using crowded public transport systems [54]. In Nepal, wheelchair users also reported not being able to mobilise independently unless they lived near a paved road, and otherwise had to rely on a family member being able to take them to a health centre [55].

Having arrived at healthcare services, people with disabilities often experience difficulties with the physical infrastructure. A review of access to healthcare services among people with disabilities in South Asia found that infrastructure and equipment did not meet their needs, including buildings, waiting rooms and examination tables [56]. Another important concern is that people with disabilities often experience difficulties paying for services [45,53,55], and studies in Tanzania, Turkey, and Thailand showed that households of people with disabilities are more likely to experience catastrophic health expenditure [57,58,59]. In Pakistan, 62% of men with disability and 87% of women with disability were financially dependent on their families, which further reduces their autonomy to seek healthcare [56]. Furthermore, negative attitudes of healthcare workers towards people with disabilities can affect a person’s ability to engage with healthcare [54,60].

These problems in access to healthcare also occur among people with NCD-related disabilities. In a qualitative study in South Africa, people who had experienced stroke reported difficulty accessing healthcare services due to lack of transport, difficulties using taxis for wheelchair-users, and difficulties mobilising from the taxi rank to the hospital; train stations were inaccessible due to stairs [61]. There is very little published on experiences of accessing healthcare among people with dementia in LMIC, but evidence from high-income settings suggests that people with dementia are less likely to receive the same quality of care as people without dementia, such as receiving screening for complications of diabetes. Also, healthcare professionals lacked skills and confidence to be able to tailor care to the needs of people with dementia, such as using visual acuity charts suited to people with cognitive impairment [62,63].

## 4. What Should Be Done about the Link between NCDs and Disability?

Despite this strong and bidirectional relationship between NCDs and disability, this association is largely ignored by the global health community. For instance, global strategies to combat NCDs are focused almost entirely on prevention and reduction of mortality [64]. The WHO Global Action Plan on NCDs from 2013 listed 9 targets and 25 indicators in relation to NCDs: none of these included any focus on disability or rehabilitation, even though the Action Plan recognises that fewer than 40% of low income countries and 60% of lower-middle income countries have any funding available for rehabilitation services [5]. The WHO’s Global Status Report on NCDs mentions disability only once in the 302 page report and rehabilitation 3 times [5].

While prevention of NCDs should of course be an important element of any proposed strategy, the needs of those already living with NCD-related disability and those who will develop NCD-related disability should not be ignored. Therefore, rehabilitation services must be integral to any NCD strategy. Furthermore, unless NCD prevention and treatment strategies are explicitly designed to be accessible and inclusive to all people with disabilities, they will miss a substantial and high-risk population.

### 4.1. Improving Rehabilitation for People with NCD-Related Disabilities

Rehabilitation services include all those that aim to “optimise functioning and reduce disability in individuals with health conditions in interaction with their environment” [65], and therefore have a wide scope and relevance for people living with NCDs. For people who have experienced stroke, rehabilitation may include increasing physical strength to assist with mobility and balance, use of assistive devices such as walking frames, speech and language therapy to aid communication, and psychological support [66]. Rehabilitation in dementia can include cognitive training as well use of compensatory strategies to achieve person-centred goals of maintaining or improving function [67]. There is good evidence that physical rehabilitation improves function, including following stroke [68,69] and that cognitive rehabilitation in dementia (i.e., on improving functioning in an everyday setting, rather than improving cognitive function per se) could improve quality of life, memory, and ability to carry out activities of daily living [70]. 

Data from LMIC are sparse, but from available literature there is a profound need for rehabilitation among different user groups with NCDS, yet insufficient services, which left many people with substantial unmet needs for rehabilitation [71]. The main reasons for the unmet needs for rehabilitation were the absence of or unequal geographical distribution of services within a country, lack of transportation, and unaffordability of the services. New initiatives are therefore needed to scale up rehabilitation services. Rehabilitation 2030 is a WHO programme to improve the availability of accessible and affordable rehabilitation, including through increasing the availability of specialist healthcare professionals. Task-sharing may also be helpful. For instance, family-led rehabilitation may be a useful for people surviving from stroke [72]. Innovation will be necessary, however, such as the use of new technology to close healthcare gaps. In India, an mHealth rehabilitation intervention involving smartphones with information about stroke and ways to manage post-stroke disability was shown to be both feasible and acceptable [73], and is being rolled out in a randomised controlled trial. A short task-based training intervention for post-stroke rehabilitation in South Africa which involved six 1-hour rehabilitation sessions over 12 weeks led to significantly improved mobility and balance in the intervention group compared to the control group [69]. 

Rehabilitation will need to extend beyond medical rehabilitation, as people with disabilities fall behind in terms of poverty, livelihoods and social inclusion [1]. The WHO recommend community-based rehabilitation in LMIC in order to improve health, education, livelihood, social inclusion and empowerment outcomes for people with disabilities [74]. There are also specific occupational rehabilitation programmes to encourage return to work for people with disabilities [75]. Regarding stroke in particular, a trial of a personalised vocational rehabilitation programme in South Africa showed that participants who received the intervention were five times more likely to return to work by 6 months than controls [76]. A systematic review on community-based rehabilitation in LMIC demonstrated that these interventions appear broadly beneficial, but also highlighted a dearth of evidence from much of the world [77]. 

### 4.2. Improving Access to Healthcare Services for People with Disabilities

The rights of people with disabilities are enshrined in the Convention on the Rights of Persons with Disabilities [78]. Clearly, people with disabilities have the same right to access to healthcare as people without disabilities, and it is crucial that the current inequity is addressed. Despite this, we found no studies evaluating how best to ensure NCD services are accessible to all. 

However, while not specifically designed to improve accessibility for people with disabilities, in multiple LMIC settings community-health workers have been tasked with aspects of prevention and management of NCDs [79,80,81], with broadly positive outcomes regarding risk factor reduction [82]. This approach may overcome some barriers to accessing healthcare that people with disabilities face, by reducing the need to travel to healthcare settings, and bringing a health promotion message into the community.

In other healthcare domains, interventions are being trialed to improve disability accessibility. In India, for example, there is ongoing work on Inclusive Eye Health aiming to use a sustainable, scalable, system-strengthening approach, which uses disaggregated disability data to inform strategy, regular training of healthcare workers on accessibility, and use of a referral network to better understand the needs of people with disabilities [83]. Yet it is clear that more evidence is needed in this area so that we can achieve Universal Health Care and avoid unnecessary NCDs among people with disabilities.

## 5. Conclusions

NCDs and disability are closely linked, and with an increasing shift in disease burden towards NCDs as well as an ageing global population, both are likely to become increasingly common over the coming decades. While focusing on prevention of NCDs to reduce premature mortality is crucial, we must ensure that NCD services are designed to be inclusive of people with disabilities and we must embed rehabilitation services within any NCD strategy.

## Figures and Tables

**Figure 1 ijerph-16-03488-f001:**
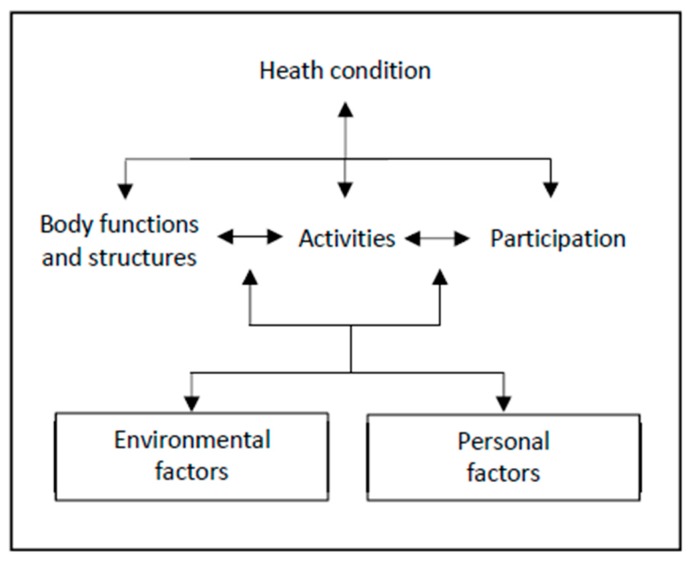
World Health Organisation International Classification of Functioning (ICF) [1].

**Figure 2 ijerph-16-03488-f002:**
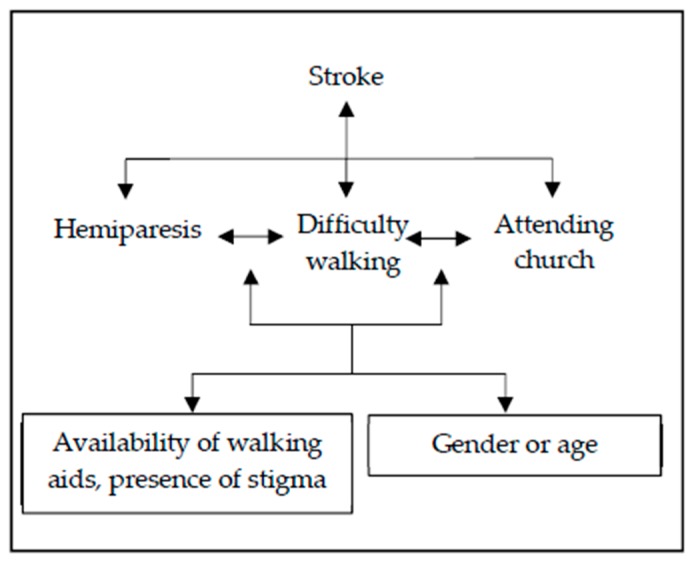
World Health Organisation International Classification of Functioning: a case study of stroke.

**Figure 3 ijerph-16-03488-f003:**
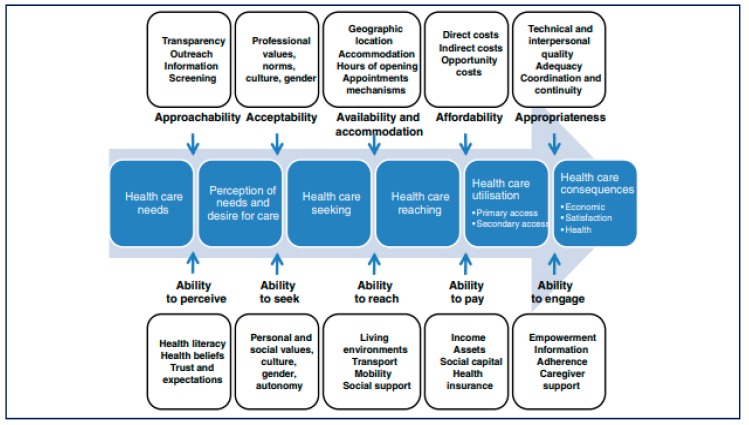
A framework of access to care conceptualised by Levesque et al [52].

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
