# Peer review of "Perspectives on Disability and Non-Communicable Diseases in Low- and Middle-Income Countries, with a Focus on Stroke and Dementia"

_ijerph, 2019, doi:10.3390/ijerph16183488_

Round 1

Reviewer 1 Report

This is a useful and timely narrative review of the literature on NCDs and disability. The author/s use stroke and dementia as case examples to illustrate some of the difficulties encountered by those who live with an NCD health condition and those who have a disability and also an NCD as either consequence or cause of disability.

The ms is easy to read, well-structured, and informative based on global literature for example, burden of disease and on local literature where this is available about particular regions and/ or nations. The authors appropriately take stroke and dementia as their case examples: these are two areas where there are some papers about NCDs and disability in LMIC. The paper makes the case for an association between NCDs and disability; and that people with disability who may be at higher risk of developing NCDs are less likely to access screening services. The paper also makes the case drawing on evidence about the beneficial impact of rehabilitation, for making rehabilitation available for those at risk of NCDs or secondary complications/ morbidity as well as those who are living with NCD.

Given the current state of the literature, the paper is strongest on the topic of people with disability, high risk of NCD and living with NCD. Where the paper provides less information is in relation to people with NCDs who may not consider themselves/ or be considered by others to be disabled.  There is some attention to the siloed approach to NCD separately to disability and vice versa. This section however could be strengthened with more information on the likelihood of persons living with NCDs also being disabled.

A significant ommission is any reference to method. The author/s clearly state in lines 44-46 that their aim is to provide a narrative review with case illustrations. This is acceptable and worthwhile. However with no attention to method, it is unclear the extent of the literature accessed, over what period of time, any exclusion criteria, and what criteria were used to include studies, and importantly the decision making about which studies to mention. The absence of method does not align with the title in which the phrase narrative review is included. The impression that the reader is left with is that the paper is more like a opinion piece, a perspective on this topic, and with references to missing items in the current 2013 WHO Global Action Plan for the prevention and control of noncommunicable diseases, reads somewhat of a 'message' to NCD planners, managers and personnel to think about and include people with disabilities in their programs.

The paper urgently requires a method section before consideration for publication. If this major ommission did not exist my recommendations would be accept with minor revisions ( a document has been provided to the editors).

Author Response

Dear Reviewer,

Many thanks for your thoughtful comments on this paper. We have taken the comments on board and amended the submission accordingly.

We appreciate your comment that without a methods section this submission should be considered to be more of an opinion piece, and thus have re-worded the title to “Perspectives on disability and non-communicable diseases in low- and middle-income countries, with a focus on stroke and dementia”.  We have also removed any mention of this being a narrative review. We hope that these changes clarify our intention and the content of the submission.

We have also addressed your comments within the text with track changes in the attached submission.

Yours sincerely,

Josephine Prynn

Reviewer 2 Report

Dear Authors,

Your narrative review on disability in relation to NCDs, in particular on stroke and dementia has a significant contribution very low in the field and I have some comments that authors should address before publication in the Journal.

Introduction

Line 33: I suggest to eliminate “animals”

Line 34:I suggest to insert multiple sclerosis

Line 37: You wrote mortality in general and then you wrote maternal and child mortality. I suggest to choose to write on mortality or in general or in details.  If you decide to write on mortality in detail I suggest to write the % for each category ( child, adult, ecc)

Line 58: I suggest to eliminate “before we” and to write “to consider”

Line 59: I suggest to eliminate “conceptualises” and to write “considers”

Line 61: I suggest to eliminate “someone” and to write “person”

Line 63: You wrote “participation restriction”. I suggest to describe what participation, social or work or both?

Line 63: You wrote “personal and environmental factors” I suggest to explain these factors

Figure 1:  disorder: I suggest to explain what disorder

                  Environmental and personal I suggest to add example in box

Line 178: I suggest to insert the study of Scaratti et al 2018

Line 232: I suggest to add a part on occupational rehabilitation

I suggest to add a paragraph on reintegration services in labour market in particular for people after stroke or MCI

Author Response

Dear Reviewer,

Many thanks for your thoughtful comments on this paper. We have taken the comments on board and amended the submission accordingly.

Below, I outline our response to each comment individually:

Line 33: I suggest to eliminate “animals”

Done

Line 34:I suggest to insert multiple sclerosis

Done

Line 37: You wrote mortality in general and then you wrote maternal and child mortality. I suggest to choose to write on mortality or in general or in details.  If you decide to write on mortality in detail I suggest to write the % for each category ( child, adult, ecc)

We are trying to emphasise two areas of global public health success in recent decades: death from communicable diseases, and maternal and child mortality. We have changed the structure of the sentence to better reflect that.

Line 58: I suggest to eliminate “before we” and to write “to consider”

Done

Line 59: I suggest to eliminate “conceptualises” and to write “considers”

Done

Line 61: I suggest to eliminate “someone” and to write “person”

Done

Line 63: You wrote “participation restriction”. I suggest to describe what participation, social or work or both? “

We have clarified this with “participation restriction, including in work, social, and family life”

Line 63: You wrote “personal and environmental factors” I suggest to explain these factors

We have clarified this – see Line 66

Figure 1:  disorder: I suggest to explain what disorder 

As the term disorder is not clear, we have removed it and replaced with the more general “health condition”

Environmental and personal I suggest to add example in box

We have added Figure 2 to work through the stroke example using the ICF framework

Line 178: I suggest to insert the study of Scaratti et al 2018

I regret that I am not completely clear which study this is referring to that is relevant to access to healthcare. I would be happy to include this reference if the reviewer can share the full citation.

Line 232: I suggest to add a part on occupational rehabilitation

Done

I suggest to add a paragraph on reintegration services in labour market in particular for people after stroke or MCI

We have added a reference to a trial on vocational rehabilitation following stroke in South Africa

Yours sincerely,

Dr Josephine Prynn

Round 2

Reviewer 1 Report

I have reviewed the revised manuscript and the author's response to my concerns. I find that the manuscript is now acceptable for publication and that the authors have addressed my concerns satisfactorily.

Reviewer 2 Report

Dear Authors,

I think your paper has improved. For me now the paper has a good structure.

For reference to add in line 178 is  Scaratti et al.,
Mapping European Welfare Models: State of the Art of Strategies for
Professional Integration and Reintegration of Persons with Chronic Diseases
 in International Journal of Environmental Research and Public Health 
15(4):781 · April 2018

Best regards